# Cognitive Profiles in the WISC-V of Children with ADHD and Specific Learning Disorders

**Angelika Becker** [1,2], **Monika Daseking** [1,2] and **Julia Kerner auch Koerner** [1,2,*]

[1] Educational Psychology, Helmut-Schmidt-University Hamburg, 22043 Hamburg, Germany; angelika.becker@hsu-hh.de (A.B.); m.daseking@hsu-hh.de (M.D.)

[2] IDeA Research Center for Individual Development and Adaptive Education of Children at Risk, 60323 Frankfurt am Main, Germany

* Correspondence: KernerAuchKoerner@hsu-hh.de

**Abstract:** Attention Deficit Hyperactivity Disorder (ADHD) has a high comorbidity with specific learning disorders (SLD). Children with ADHD and children with SLD show specific cognitive deficits. This study aims to examine similarities and differences between cognitive profiles of children with ADHD + SLD, children with SLD only, and a control group to find out whether specific or shared deficits can be identified for the groups. We compared the WISC-V profiles of 62 children with ADHD and SLD (19 girls, *M*-age = 10.44; *SD* = 2.44), 35 children with SLD (13 girls, *M*-age = 10.21; *SD* = 2.11) and 62 control children without ADHD or SLD (19 girls, *M*-age = 10.42; *SD* = 2.39). The ADHD + SLD group performed worse than the control group in the WISC-V indices WMI, PSI, FSIQ, AWMI, CPI and worse than the SLD group in these indices and the VCI, NVI and GAI. Therefore, compared to children with SLD, children with ADHD + SLD did not show specific impairments in any particular cognitive domain but rather non-specific impairment in almost all indices. Hence, the WISC-V is suited to depict the cognitive strength and weaknesses of an individual child as a basis for targeted intervention.

**Keywords:** ADHD; specific learning disorders; WISC-V; arithmetical disorder; dyscalculia; spelling disorder; reading disorder; dyslexia



## 1. Introduction

With a prevalence of about 5%, Attention Deficit Hyperactivity Disorder (ADHD) is one of the most common mental disorders in childhood [1,2]. The number of children with specific learning disorders (SLD; difficulties in learning to read, write or calculate) varies in a similar high range (2–8%, [3–5]). The comorbidity between the two disorders is substantial. Approximately 20–70% of children with a clinical diagnosis of ADHD also suffer from SLD [6–9]. Vice versa, about 20–28% of children with SLD also show clinical levels of ADHD symptoms [10,11]. The substantial overlap between the two disorders calls into question whether both share similar cognitive deficits or whether the cognitive deficits are specific to the disorders.

### 1.1. Cognitive Profiles of Children with ADHD

Children with ADHD are characterized by age-inappropriate symptoms of inattention, hyperactivity, and impulsivity [12,13]. Furthermore, they also show consistent impairments in other areas of cognitive functioning such as executive functions [14], working memory [15,16], and processing speed [17]. A meta-meta-analysis of 34 meta-analyses of cognitive profiles in ADHD (all ages) found moderate impairments in the following domains: working memory, reaction time variability, response inhibition, intelligence/achievement, and planning/organization [18]. Theories of ADHD suggest a core deficit in behavioral inhibition (i.e., inhibition of a prepotent response, stopping an ongoing response or interference control) which then affects the other cognitive domains [19].

### 1.2. Cognitive Profiles of Children with Specific Learning Disorders

Children with SLD also show weaknesses in working memory [20] and processing speed [17]. Some suggest that in addition to the deficits just mentioned there are also specific impairments for different types of SLD [21–23]. Children with reading disorders (i.e., dyslexia) show deficits in tasks requiring phonological processing [22,23]. In contrary, children with difficulties in the acquisition of arithmetical skills (i.e., dyscalculia) have deficits in tasks with requirements for visual-spatial skills [21]. However, these findings of specific impairments for different SLDs are not as consistent as the finding of impaired working memory and processing speed in children with SLD in general.

### 1.3. Measuring Cognitive Profiles in Children with Specific Disabilities Using IQ Tests

Cognitive profiles and specific cognitive deficits in particular can be displayed using intelligence tests like the Wechsler Intelligence Scale for Children (WISC-V, [24]). The WISC-V differs from the previous version of Wechsler intelligence tests, the WISC-IV [25] in several points. The underlying intelligence structure was changed from a four-factor to a five-factor-model that should be adequately represented by the factor structure of the test (see [24], technical manual). The WISC-V includes five primary indices (Verbal Comprehension Index [VCI], Visual Spatial Index [VSI], Fluid Reasoning Index [FRI], Working Memory Index [WMI], Processing Speed Index [PSI]), five ancillary indices (Quantitative Reasoning Index [QRI] Auditory Working Memory Index [AWMI] Nonverbal Index [NVI], General Ability Index [GAI], Cognitive Proficiency Index [CPI]) and the Full Scale IQ (FSIQ). On the basis of confirmatory factorial analyses, the Perceptual Reasoning Index was split into the new VSI and the FRI. New subtests were added to the test (Figure Weights, Visual Puzzles and Picture Span). Now only seven subtests contribute to the FSIQ instead of 10 as in the WISC-IV. The primary WMI now also includes a visual subtest (Picture Span) and an auditory subtest (Digit Span) instead of only auditory subtests as in the WISC-IV, where the WMI was composed of the subtests Digit Span and Letter-Number Sequencing. In the WISC-V these two subtests (Digit Span and Letter-Number Sequencing) are now combined in the ancillary AWMI.

Test batteries such as the WISC-V allow to depict the cognitive profile of children in different domains (i.e., verbal comprehension, visual spatial, fluid reasoning, working memory, processing speed) in a single test whose standard values are based on the same standardization sample. Results in different domains are therefore comparable, as opposed to using different tests (based on different standardization samples) to evaluate deficits in different domains. Using a test battery can help to detect strengths and weaknesses that could not emerge when a unitary IQ value is considered [26].

Furthermore, in the diagnostic process of ADHD a comprehensive IQ diagnostic plays an important role for ruling out being over- or underchallenged as alternative explanations for hyperactivity, impulsivity, and inattention especially in the school context.

For the diagnosis of SLD ICD-10 [13] and ICD-11 [27] require a discrepancy of at least one standard deviation between specific learning achievement and IQ. The Diagnostic and Statistical Manual of Mental Disorders (DSM-5 [12]) no longer requires a discrepancy to IQ, but intellectual disability must still be excluded. Therefore, an IQ test is central in the diagnosis of SLD, as well as in the diagnosis of ADHD [28].

Previous studies using the WISC-II, III, or IV have shown that children with ADHD consistently show weaker performances on tasks related to working memory or processing speed [29–34]. When using the WISC to assess children with SLD, they show the most consistent deficits in working memory [21,35,36]. However, some also find differences in processing speed [21]. Children with ADHD and language-based SLD have even poorer working memory than children with ADHD only [37].

Meta-analyses also shows lower overall intellectual abilities for children with ADHD compared to healthy controls [18,38]. Some of the differences might be accounted for by lower working memory or processing speed but parts might also be due to inattentive or impulsive behavior during testing. The assumption is that children with ADHD do

not show lower intellectual abilities than children without ADHD per se (see [39] for a detailed discussion).

*1.4. Aims*

The present study aims to examine similarities and differences between cognitive profiles of children with ADHD + SLD, children with SLD, and a control group of healthy children to find out whether specific and/or shared deficits can be identified for the groups. Although, studies have compared the WISC profiles of children with ADHD and children with SLD [40] research using the WISC-V is sparse.

Based on previous research we expect that the group with ADHD and SLD shows deficits in working memory and processing speed but not in the fluid reasoning and general ability indices. The group with SLD and no ADHD should show deficits in working memory.

## 2. Methods

Sixty-two children and adolescents ($M$-age = 10.43 years, $SD$ = 2.43; 19 female = 30.6%) with a diagnosis of both ADHD and comorbid learning disorders were recruited in co-operation with a counseling center specialized in learning disorders and a children and youth psychiatrist. Demographic data of our sample are displayed in Table 1. The diagnosis of ADHD was tested and diagnosed by the treating psychiatrist with standardized tests (KITAP; [41]; TAP; [42], parental report (CBCL/6-18R, [43]; FBB-ADHS from the DISYPS-III, [44]), self-report (YSR/6-18R, [43]; SBB-ADHS from DISYPS-III, [44]), and if available, teachers report (TRF/6-18R, [43], FBB-ADHS from the DISYPS-III, [44]) as well as clinical interviews and behavior observations. Only children with ADHD (F90.0), not with ADD (F98.8) were included in this sample. Children with an overall-IQ < 70 and severe neurological or psychological problems were excluded from the sample ($n$ = 4). Assessment of intelligence was done at the psychology laboratory of our institution by experienced psychologists using the German version of the WISC-V [45]. In a second session, standardized German school achievement tests for reading, spelling, and arithmetic were conducted with the participants. Reading achievement was measured using the ELFE II [46] and the LGVT 6–12 [47]. Spelling was tested with the HSP 1–10 [48] and the DRT 1 [49], WRT 2+ [50], 3+ [51] or 4+ [52]. To assess achievement in arithmetic we used the HRT 1–4 [53] and the DEMAT 2+ [54], 3+ [55], 4 [56], 5+ [57] or 6+ [58]. Scores below a $t$-score of 40 in one of the tests were here classified as learning disorder (SLD).

**Table 1.** Demographic description of the sample by group.

| | ADHD + SLD Group ($n$ = 62) | SLD Group ($n$ = 35) | CONTROL ($n$ = 62) | ADHD ($n$ = 13) | Group Differences |
|---|---|---|---|---|---|
| sex ($n$ and % female) | 19 (30.6%) | 13 (37.1%) | 19 (30.6%) | 3 (23.1%) | 0.804 [a] |
| $M$ age (SD) | 10.44 (2.44) | 10.21 (2.11) | 10.42 (2.39) | 10.53 (2.48) | 0.971 [b] |
| Type of school | | | | | 0.241 [a] |
| Primary school | 38 (62.3%) | 22 (64.7%) | 37 (59.7%) | 7 (53.8%) | |
| Secondary school, graduation after 9th grade (German: Hauptschule) | 2 (3.3%) | 2 (5.9%) | 4 (6.5%) | — | |
| Grammar school, graduation after 12th or 13th grade, univerity entrance degree (German: Gymnasium) | 4 (6.6%) | 3 (8.8%) | 4 (6.6%) | 5 (38.5%) | |
| Comprehensive school, different kinds of degrees can be obtained after 9th or 12th/13th grade (German: Gesamtschule) | 14 (23%) | 6 (17.6%) | 15 (24.2%) | 1 (7.7%) | |
| Special school (German: Förderschule) | 3 (4.9%) | 1 (2.9%) | 2 (3.2%) | — | |

**Table 1.** *Cont.*

| | ADHD + SLD Group (*n* = 62) | SLD Group (*n* = 35) | CONTROL (*n* = 62) | ADHD (*n* = 13) | Group Differences |
|---|---|---|---|---|---|
| Parental education | | | | | 0.451 [a] |
| Lower education level | 10 (16.1%) | 3 (8.6%) | 8 (12.9%) | – | |
| Medium education level | 26 (41.9%) | 16 (45.7%) | 26 (41.9%) | 2 (15.4%) | |
| High education level | 10 (16.1%) | 7 (20%) | 11 (17.1%) | 4 (30.8%) | |
| Highest education level | 16 (25.8%) | 8 (22.9%) | 16 (25.8%) | 7 (53.8%) | |

[a] *p*-value of $X^2$. [b] *p*–value of $X^2$ retrieved from Kruskall-Wallis. Note. Parental education is defined as the highest level of education achieved by either one parent or both (low educational level = no diploma or school certificate after 9th grade, medium educational level = school certificate after 10th grade, high educational level = university entrance qualification/certificate after 12th or 13th grade, and highest educational level = college/university degree).

A group of *n* = 35 children that was diagnosed with SLD, but not with ADHD was also included in this sample (see Table 1 for an overview). Cases were classified according to the ICD-10 coding system [13] as specific reading disorder (F81.0), specific spelling disorder (F81.1), specific disorder of arithmetical skills (F81.2) or mixed disorder of the scholastic skills (F81.3) (for an overview see Table 2). A small number of children was diagnosed with ADHD but not with SLD (*n* = 13). Due to this small sample size, we did not include this group in our hypotheses, but we did perform exploratory non-parametric group comparisons and included them in the Appendix.

**Table 2.** Number of children classified to different types of learning disorders.

| | Specific Reading Disorder (*N*) | Specific Spelling Disorder (*N*) | Specific Disorder of Arithmetical Skills (*N*) | Mixed Disorder of the Scholastic Skills (*N*) |
|---|---|---|---|---|
| ADHD &SLD | 19 | 16 | 3 | 24 |
| SLD | 11 | 14 | 2 | 8 |

A control sample was formed by selecting children from the German WISC-V standardization sample that matched the ADHD + SLD group by age, sex, parental educational level and type of school (*n* = 62) (see Table 1). Parental report was given that the children from the control group (CONTROL) did not suffer from ADHD/ADD or any learning disorders. Prior to testing, parental informed consent was given.

The total number of the tested sample thus is *N* = 172 (ADHD + SLD = 62, CONTROL = 62, SLD = 35, ADHD = 13).

## 3. Results

### 3.1. Statistical Analyses

Statistical analysis was performed in SPSS [59] using a MANOVA with the independent variable group (ADHD + SLD, SLD, CONTROL) and the five primary and five ancillary indices of the WISC-V as dependent variables. If significant interactions were observed, post-hoc t-tests were conducted. The statistical alpha level was set below.05. Eta square was calculated as effect size for parametrical group comparisons and Cohen's *d* effect size was calculated for comparisons of two groups. Effect sizes were classified according to Cohen [60] as small effects ($\eta^2 = 0.01$; *d* = 0.20), moderate effects ($\eta^2 = 0.06$; *d* = 0.50), and large effects ($\eta^2 = 0.14$; *d* = 0.80). To explore the ADHD + SLD group further, in a second step the ADHD + SLD group was divided into children that had problems in reading, writing or both but no problems in arithmetic (*n* = 35), and children that had problems in arithmetic (with or without problems in reading/and or writing) (*n* = 27). Here, also a MANOVA was calculated with the between subject factor group (with problems in arithmetic versus with no problems in arithmetic) and the five primary and five ancillary indices of the WISC-V as dependent variables. See the Appendix A for non-parametric group comparisons of the five primary and five ancillary index scores of the WISC-V (Kruskal-Wallis and Mann-Whitney-U) with the small ADHD only sample included. Effect

sizes of the non-parametrical computations were classified according to Cohen [60] as small effect *r* = 0.1, moderate effect, *r* = 0.3 or large effect, *r* = 0.5.

The scaled scores of the subtests contributing to the primary index Working Memory and the ancillary index Auditory Working Memory were compared by non-parametrical tests for all groups (Kruskal–Wallis and Mann–Whitney-U–test) (see Appendix).

### 3.2. Descriptive Statistics and Group Comparison

Descriptive statistics of the scores for all primary and ancillary WISC-V indices and their comparisons across groups are reported in Table 3. The ADHD + SLD group had lower scores in all indices, see also Figure 1 for primary WISC-V index scores and FSIQ und Figure 2 for ancillary WISC-V index scores (see Appendix B for the figures with the ADHD only group included).

**Table 3.** Mean and standard deviation for all primary and ancillary WISC-V indices by group.

| WISC-V Index | ADHD + SLD n = 62 | | SLD n = 35 | | CONTROL n = 62 | | MANOVA | | | Post Hoc (See also Appendix C) |
|---|---|---|---|---|---|---|---|---|---|---|
| | **M** | **SD** | **M** | **SD** | **M** | **SD** | **F(df1/df2)** | **p** | **η²** | |
| VCI | 97.47 | 14.37 | 105.41 | 11.29 | 100.76 | 11.53 | 4.338 | (2/155) | 0.015 | 0.05 | ADHD + SLD < SLD |
| VSI | 99.18 | 13.63 | 102.32 | 13.92 | 101.00 | 14.44 | 0.602 | (2/155) | 0.549 | 0.01 | – |
| FRI | 97.90 | 13.98 | 103.03 | 13.08 | 101.74 | 13.34 | 2.016 | (2/155) | 0.137 | 0.03 | – |
| WMI | 93.48 | 13.28 | 101.65 | 11.55 | 102.27 | 14.16 | 7.845 | (2/155) | 0.001 | 0.09 | ADHD + SLD < CONTROL; ADHD + SLD < SLD |
| PSI | 90.48 | 12.30 | 97.76 | 10.85 | 102.19 | 13.83 | 13.464 | (2/155) | <0.001 | 0.15 | ADHD + SLD < CONTROL; ADHD + SLD < SLD |
| FSIQ | 93.87 | 12.98 | 102.59 | 12.57 | 101.63 | 12.30 | 7.805 | (2/155) | 0.001 | 0.09 | ADHD + SLD < CONTROL; ADHD + SLD < SLD |
| QRI | 95.35 | 13.25 | 101.56 | 11.66 | 99.84 | 13.56 | 3.053 | (2/155) | 0.050 | 0.04 | – |
| AWMI | 87.98 | 10.52 | 94.88 | 12.31 | 101.81 | 13.17 | 20.560 | (2/155) | <0.001 | 0.21 | ADHD + SLD < CONTROL; ADHD + SLD < SLD; SLD < CONTROL |
| NVI | 96.85 | 14.03 | 103.29 | 12.22 | 101.89 | 14.35 | 3.142 | (2/155) | 0.046 | 0.04 | ADHD + SLD < SLD |
| GAI | 97.32 | 14.07 | 104.79 | 12.28 | 101.52 | 12.11 | 3.925 | (2/155) | 0.022 | 0.05 | ADHD + SLD < SLD |
| CPI | 90.02 | 12.64 | 99.82 | 10.54 | 102.76 | 13.82 | 16.544 | (2/155) | <0.001 | 0.18 | ADHD + SLD < CONTROL; ADHD + SLD < SLD |

VCI, Verbal Comprehension Index; VSI, Visual Spatial Index; FRI, Fluid Reasoning Index; WMI, Working Memory Index; PSI, Processing Speed Index; FSIQ, Full Scale IQ; QRI, Quantitative Reasoning Index; AWMI, Auditory Working Memory Index; NVI, Nonverbal Index; GAI, General Ability Index; CPI, Cognitive Proficiency Index.

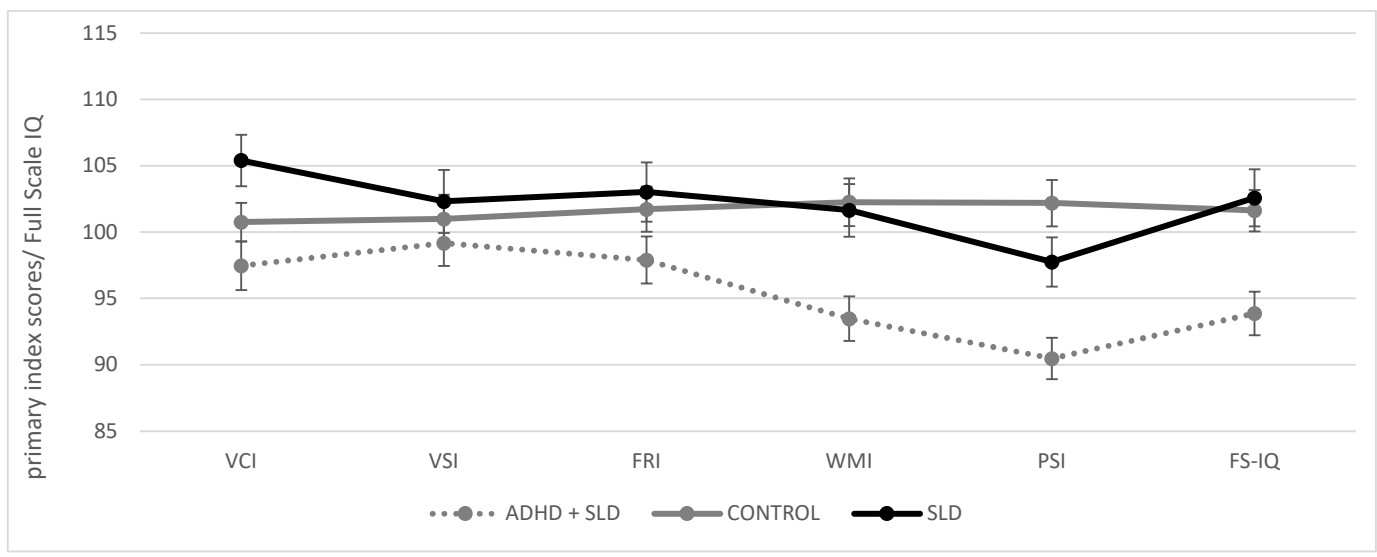

**Figure 1.** Mean scores and standard error of the primary WISC-V index scores and the FSIQ as comparison between groups. Note: VCI, Verbal Comprehension; VSI, Visual Spatial Index; FR, Fluid Reasoning Index; WMI, Working Memory Index; PSI, Processing Speed Index; FSIQ, Full Scale IQ.

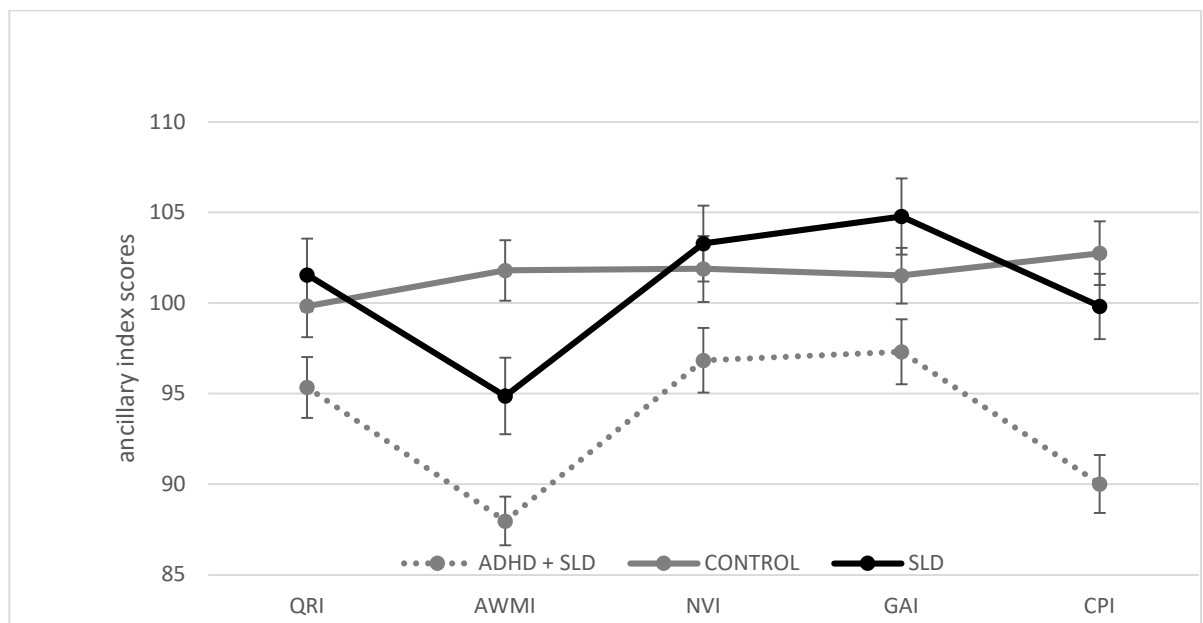

**Figure 2.** Mean scores and standard error of the ancillary WISC-V index scores as comparison between groups. Note: QRI, Quantitative Reasoning Index; AWMI, Auditory Working Memory Index; NVI, Nonverbal Index; GAI, General Ability Index; CPI, Cognitive Proficiency Index.

Significant group differences were seen in eight indices, namely in the VCI, the WMI, the PSI, the FSIQ, the AWMI, the NVI, the GAI, and the CPI. A large effect according to Cohen (1988) was seen in the PSI, the AWMI and the CPI [60].

Post hoc *t*-tests for all scales with significant MANOVA effects revealed that the ADHD + SLD group differed from the control group (CONTROL) in five indices WMI, PSI, FSIQ, AWMI, CPI and in eight indices from the SLD group VCI, WMI, PSI, FSIQ, AWMI, NVI, GAI, CPI. The detailed results for the post hoc tests are displayed in Appendix B. The SLD group differed only in the AWMI from CONTROL (see Appendix C).

To test if children and adolescents with ADHD + SLD and problems in arithmetic differ from children and adolescents with ADHD + SLD and no problems in arithmetic, different index scores between those groups were compared (see Table 4). The group with problems in arithmetic showed significant lower mean scores in almost all indices (except for PSI and VSI, see Table 4).

**Table 4.** Group comparisons between children with ADHD + SLD with and without problems in arithmetic.

| WISC-V Index | ADHD + SLD with Problems in Arithmetic *n* = 27 | | ADHD + SLD without Problems in Arithmetic *n* = 35 | | ANOVA | | | |
|---|---|---|---|---|---|---|---|---|
| | **M** | **SD** | **M** | **SD** | **F(df1/df2)** | | **p** | **$\eta^2$** |
| VCI | 93.11 | 14.62 | 100.83 | 13.41 | 4.663 | (1/60) | 0.035 | 0.07 |
| VSI | 95.44 | 15.21 | 102.60 | 11.69 | 3.750 | (1/60) | 0.058 | 0.06 |
| FRI | 91.85 | 12.93 | 102.57 | 12.73 | 10.658 | (1/60) | 0.002 | 0.15 |
| WMI | 88.96 | 13.76 | 96.97 | 12.12 | 5.992 | (1/60) | 0.017 | 0.09 |
| PSI | 88.33 | 13.16 | 92.14 | 11.51 | 1.473 | (1/60) | 0.230 | 0.02 |
| FSIQ | 88.81 | 12.93 | 97.77 | 11.76 | 8.103 | (1/60) | 0.006 | 0.12 |
| QRI | 88.96 | 10.59 | 100.29 | 13.11 | 13.381 | (1/60) | 0.001 | 0.18 |
| AWMI | 84.04 | 10.77 | 91.03 | 09.37 | 7.448 | (1/60) | 0.008 | 0.11 |
| NVI | 91.04 | 14.75 | 101.34 | 11.79 | 9.351 | (1/60) | 0.003 | 0.14 |
| GAI | 91.96 | 14.60 | 101.46 | 12.31 | 7.707 | (1/60) | 0.007 | 0.11 |
| CPI | 86.15 | 12.93 | 93.00 | 11.73 | 4.758 | (1/60) | 0.033 | 0.07 |

VCI, Verbal Comprehension Index; VSI, Visual Spatial Index; FRI, Fluid Reasoning Index; WMI, Working Memory Index; PSI, Processing Speed Index; FSIQ, Full Scale IQ; QRI, Quantitative Reasoning Index; AWMI, Auditory Working Memory Index; NVI, Nonverbal Index; GAI, General Ability Index; CPI, Cognitive Proficiency Index.

## 4. Discussion

No group differences emerged in the VSI, in the FRI and in the QRI. In all other primary and ancillary indices, the ADHD + SLD group performed worse than the CONTROL group and/or the SLD group. Post hoc tests indicated that the ADHD + SLD group performed worse than the CONTROL group in WMI, PSI, FSIQ, AWMI, CPI with large effect sizes and worse than the SLD group in VCI, WMI, PSI, FSIQ, AWMI, NVI, GAI, CPI with large effect sizes (except for NVI with a medium effect).

### 4.1. Comparison of ADHD + SLD and CONTROL

Therefore, in accordance with our expectations compared to a matched control group, children with ADHD and SLD did show deficits in working memory (WMI and AWMI) and processing speed (PSI and CPI). Also, they did not show impairments in the FRI and the GAI. Furthermore, the ADHD + SLD group differed from the CONTROL in FSIQ, which is not surprising since it takes into account subtests from WMI and PSI.

### 4.2. Comparison of ADHD + SLD and SLD

Compared to the SLD group, the group of children with SLD and ADHD did not show specific impairments in any particular cognitive domain but rather non-specific impairments in almost all indices of the WISC-V.

Our clinical group with SLD showed surprisingly little cognitive impairment, and only the secondary AWMI differed from the control group. However, this comparison is limited in its accountability because the control group was matched to the ADHD + SLD group and not to the SLD group and the SLD group has fewer cases compared to the other two groups. But no group differences emerged in sex, age, type of school, and parental education.

The finding that the SLD group is only impaired in AWMI compared to a control group seems to be in contrast to previous studies finding impairments in working memory and processing speed in children with SLD [21,28,35,36,61]. However, previous studies often used auditory working memory tests. The WMI from the WISC-IV included only auditory subtests. In contrast, the WMI from the WISC-V is based on one visual (subtest Picture Span) and one auditory subtest (subtest Digit Span). The AWMI incorporates two subtests, both of which address auditory memory performance (subtests Digit Span and Letter-Number Sequencing) and is more comparable to the WMI from the WISC-IV. Therefore, our finding that the SLD group is only impaired in auditory working memory is actually in accordance with previous results. To look at the comparison of auditory and visual working memory in more detail compared the performance of the groups in the working memory subtests as post hoc exploratory analysis (see Appendix D). The ADHD + SLD group showed the highest impairment in the auditory subtest Digit Span and Letter-Number Sequencing compared to the CONTROL. The SLD Group differed from the CONTROL group only in the subtest Letter-Number Sequencing, which is auditory and also the most complex memory span task. Therefore, the domain (auditory or visual) in which the working memory is assessed has to be kept in mind when focusing on children with SLD, since children with language-related learning disorders often have a deficit in phonological memory (like mentioned before and as also shown in our sample via the index score AWMI), but not in the visual-spatial memorization of information.

The contrast to previous research regarding processing speed could be due to our sample being a clinical utilization sample (families that contacted an advice center for SLDs) that might show less impairment compared to clinical samples. On the other hand previous studies on cognitive profiles of children with SLD did not control for ADHD [21,28,35,36,61], therefore their results of impaired working memory and processing speed in children with SLD might be due to ADHD symptoms in their samples.

To summarize, compared to the SLD group the ADHD + SLD group showed non-specific impairments with lower scores in almost all indices of the WISC-V, which could be due to an accumulation of problems that has been reported before [7,22]. Another reason

might be that the SLD group in this study is somewhat biased and performs better than one would expect this group in the different indices apart from the AWMI (see also [21,28]).

### 4.3. Exploratory Comparison of ADHD + SLD and ADHD

An exploratory comparison of children with ADHD and SLD to our very small sample of children with ADHD only ($N$ = 13) resulted in significant group differences in VCI, WMI, QRI and AWMI, with moderate effect sizes (see Appendix A, Table A2). However, in many indices the ADHD group had above average IQ values compared to the CONTROL group (see Appendix A, Table A1 and Appendix B, Figures A1 and A2). Children were sent to us if problems had occurred at school and a children and youth psychiatrist wanted them to be tested with IQ and reading, writing and arithmetic's test. Therefore, children with ADHD and no problems in reading, writing or arithmetic's might not have been included in our sample and we cannot draw any conclusions about the specificity of impairments of children with ADHD and SLD compared to children with ADHD only. Previous literature using the WISC-IV indicated that children with ADHD and language-based SLD have even poorer working memory than children with ADHD only [37]. Another study comparing children with dyslexia only and children with dyslexia and ADHD showed that children with dyslexia only mainly had deficits in the phonological loop, their language abilities and in the rapid naming task, while children with both ADHD and dyslexia mainly had problems in the central-executive working memory [62].

### 4.4. Exploratory Comparison of Children with and witout Arithmetic Problems in the ADHD + SLD Group

Children with ADHD and problems in reading, writing, and arithmetic showed lower scores in all WISC-V indices compared to children with ADHD and problems only in reading and writing. This is line with previous research showing no specific impairment for different disorders but rather an accumulation of problems [7,21,63]. There are studies that suggest that a comorbid reading disorder might occur in a subgroup of children with ADHD with more severe cognitive deficits [64–67].

Children that showed also problems with arithmetic showed lower scores in almost all indices than children with reading, spelling or mixed reading and spelling problems. This finding goes in line with previous studies showing that children with mixed disorders show a general lower intellectual profile [21,23,37]. Cognitive deficits underlying impairments in SLD might be additive [22], even more if there is a comorbid ADHD diagnosis. Unfortunately, our sample was too small to compare children with problems in arithmetic only to children with problems in reading or spelling only. However, previous findings point to both common and distinct cognitive deficits in this group of children with shared weaknesses in working memory, processing speed, and verbal comprehension [23]. Children with isolated impairment in arithmetic seem to show a more general cognitive deficit including perceptual reasoning [68,69], while children with dyslexia seem to show a specific phonological deficit [22].

### 4.5. Limitations and Directions for Future Research

We compared a group with ADHD and SLD to a group with SLD only. A comparison with a group with ADHD only would have been helpful to find out which of the deficits that we found in the combined group with ADHD and SLD derive from the ADHD symptoms. However, children with ADHD very often have also comorbid problems (20–70%, [6–9]) and children with only ADHD and no comorbidities are rare in practice. In our sample only 13 children with ADHD did not have a comorbid specific learning disorder and due to this small number, we did not include them in the analyses. Therefore, future research should address the question if children with ADHD only show specific cognitive profiles in the WISC-V compared to children with ADHD + SLD and children with SLD only. For example, a study using the WISC-IV shows that children with ADHD and language-based SLD have even poorer working memory than children with ADHD only [37].

Our sample consisted of families contacting a counseling center specialized in learning disorders or a child and adolescent psychiatric practice (i.e., a clinical utilization sample) and is therefore not representative. This might have led to an overrepresentation of children with multiple comorbid (psychiatric) problems. A replication of cognitive profiles in a representative population-based sample would be helpful. However, our study might be helpful for practitioners in specialized counseling centers, because they most commonly face the challenges of diagnosing and treating children with multiple comorbid problems.

*4.6. Implications for Practice*

Despite these limitations, these results suggest that deficits of ADHD versus SLD versus ADHD + SLD are not specific. The cognitive profiles of the WISC-V can of course not be used to confirm a diagnosis but can be very helpful to the individual profile of one child's cognitive strength and weaknesses and be a valuable base for an individualized treatment plan. There are for example different trainings for either processing speed or working memory functioning. An advantage of using the WISC-V instead of different tests for different constructs (like working memory and processing speed) is that are based on the same standardization and results can be compared directly.

Since the ADHD symptoms did lead to such an accumulation of problems over and above the SLD in our study they should be taken into account when (I) using the WISC-V and (II) treating children with SLD. A meta-analysis shows lower overall intellectual abilities for children with ADHD compared to healthy controls [38]. Some of the differences might be due to lower working memory or processing speed but clinicians often also fear that the true intellectual level will be underestimated because of the inattentive or impulsive behavior during testing. Therefore, (I) when administering tests like the WISC-V ADHD symptoms should be taken into account (see [39] for a detailed discussion). Clinicians should support the attention of the child as much as possible within the standardized procedures by repeating the instructions or redirecting the child's attentional focus back to the test material or praising the child for its cooperation and/or willingness to make an effort. Furthermore, more breaks in-between subtests might be needed when testing children with ADHD.

Furthermore, (II) our results and previous research show that children with both ADHD and SLD have more severe cognitive problems than children with SLD only [7,37]. Therefore, teachers need to be informed about the high comorbidity between the two disorders and diagnostics of both SLD and ADHD should be initialized as early as possible if problems occur. Prominent ADHD symptoms might restrict the effectiveness of learning interventions. Particularly symptoms of inattention might make it difficult for a child to focus on learning activities especially in domains they know they are not good at, like the area of their SLD. If ADHD symptoms are present these should be treated before or in parallel to the treatment of the SLD. A treatment of SLD in children with undiagnosed ADHD would be much less constructive and sustainable.

**Author Contributions:** Conceptualization, A.B. and J.K.a.K.; methodology, A.B.; formal analysis, A.B.; investigation, A.B.; resources, M.D.; data curation, A.B.; writing—original draft preparation, A.B. and J.K.a.K.; writing—review and editing, M.D.; visualization, A.B.; supervision, M.D.; project administration, A.B. All authors have read and agreed to the published version of the manuscript.

**Funding:** This research is based on health service research (Versorgungsforschung) and received no external funding.

**Institutional Review Board Statement:** The study was conducted according to the guidelines of the Declaration of Helsinki. Ethical review and approval were waived for this study because we fully informed the parents and children before participation (no deception), it was a counseling (not clinical) setting, the methods were not invasive and there were no psychopharmacological interventions. Therefore, in line with the standards of the German Research Society (Deutsche Forschungsgesellschaft; DFG), we did not apply to an ethics committee.

**Informed Consent Statement:** Parental informed consent was obtained from all subjects involved in the study.

**Data Availability Statement:** Since this research is based on health service research (Versorgungsforschung) the data are not publicly available.

**Acknowledgments:** We thank the children and their parents for participating in the study.

**Conflicts of Interest:** The authors declare no conflict of interest.

## Appendix A

Non-parametric group comparisons of the five primary and five ancillary index scores of the WISC-V (Kruskal-Wallis and Mann-Whitney-U) with the small ADHD only sample included. Effect sizes of the non-parametrical computations were classified according to Cohen [60] as small effect $r = 0.1$, moderate effect, $r = 0.3$ or large effect, $r = 0.5$.

**Table A1.** Mean and standard deviation for all primary and ancillary WISC-V indices by group and results of Kruskal-Wallis test.

| WISC-V Index | ADHD + SLD $n = 62$ | | SLD $n = 34$ | | CONTROL $n = 62$ | | ADHD $n = 13$ | | Kruskal-Wallis | | |
|---|---|---|---|---|---|---|---|---|---|---|---|
| | **M** | **SD** | **M** | **SD** | **M** | **SD** | **M** | **SD** | **$X^2$(df)** | | ***p*** [a] |
| VCI | 97.47 | 14.37 | 105.41 | 11.29 | 100.76 | 11.53 | 105.80 | 9.62 | 8.99 | (3) | 0.029 |
| VSI | 99.18 | 13.63 | 102.32 | 13.92 | 101.00 | 14.44 | 106.00 | 12.89 | 2.64 | (3) | 0.450 |
| FRI | 97.90 | 13.98 | 103.03 | 13.08 | 101.74 | 13.34 | 106.77 | 12.93 | 6.09 | (3) | 0.107 |
| WMI | 93.48 | 13.28 | 101.65 | 11.55 | 102.27 | 14.16 | 100.77 | 12.14 | 14.49 | (3) | 0.002 |
| PSI | 90.48 | 12.30 | 97.76 | 10.85 | 102.19 | 13.83 | 94.69 | 12.64 | 25.44 | (3) | <0.001 |
| FSIQ | 93.87 | 12.98 | 102.59 | 12.57 | 101.63 | 12.30 | 101.38 | 9.98 | 12.88 | (3) | 0.005 |
| QRI | 95.35 | 13.25 | 101.56 | 11.66 | 99.84 | 13.56 | 109.69 | 10.29 | 14.64 | (3) | 0.002 |
| AWMI | 87.98 | 10.52 | 94.88 | 12.31 | 101.81 | 13.17 | 95.85 | 8.92 | 32.87 | (3) | <0.001 |
| NVI | 96.85 | 14.03 | 103.29 | 12.22 | 101.89 | 14.35 | 104.85 | 11.05 | 5.38 | (3) | 0.146 |
| GAI | 97.32 | 14.07 | 104.79 | 12.28 | 101.52 | 12.11 | 105.85 | 11.87 | 7.03 | (3) | 0.071 |
| CPI | 90.02 | 12.64 | 99.82 | 10.54 | 102.76 | 13.82 | 97.00 | 11.23 | 28.95 | (3) | <0.001 |

[a] asymptotic significance level; VCI, Verbal Comprehension Index; VSI, Visual Spatial Index; FRI, Fluid Reasoning Index; WMI, Working Memory Index; PSI, Processing Speed Index; FSIQ, Full Scale IQ; QRI, Quantitative Reasoning Index; AWMI, Auditory Working Memory Index; NVI, Nonverbal Index; GAI, General Ability Index; CPI, Cognitive Proficiency Index.

**Table A2.** Non-parametrical post-hoc group comparisons by Mann–Whitney-U for scales with significant effect in the Kruskal–Wallis test.

| WISC-V Index | ADHD + SLD versus CONTROL | | | | ADHD + SLD versus SLD | | | | ADHD + SLD versus ADHD | | | | SLD versus CONTROL | | | | SLD versus ADHD | | | | ADHD versus CONTROL | | | |
|---|---|---|---|---|---|---|---|---|---|---|---|---|---|---|---|---|---|---|---|---|---|---|---|---|
| | U [a] | Z [b] | p [c] | r [d] | U | Z | p | r | U | Z | p | r | U | Z | p | r | U | Z | p | r | U | Z | p | r |
| VCI | 3575.50 | −1.501 | 0.133 | −0.135 | 2712.00 | −2.455 | **0.014** | −0.249 | 22150.00 | −1.979 | **0.048** | −0.228 | 2821.00 | −1.703 | 0.089 | −0.173 | 314.00 | −0.105 | 0.917 | −0.015 | 2268.50 | −1.230 | 0.219 | −0.142 |
| WMI | 3196.50 | −3.399 | **0.001** | −0.305 | 2673.00 | −2.751 | **0.006** | −0.279 | 2214.50 | −1.987 | **0.047** | −0.229 | 1655.50 | −0.448 | 0.654 | −0.045 | 851.00 | −0.152 | 0.880 | −0.022 | 475.00 | −0.267 | 0.790 | −0.031 |
| PSI | 2914.00 | −4.813 | **<0.001** | −0.432 | 2664.00 | −2.818 | **0.005** | −0.286 | 2291.00 | −0.913 | 0.361 | −0.105 | 1455.50 | −1.956 | 0.050 | −0.198 | 276.50 | −0.979 | 0.328 | −0.014 | 361.00 | −1.869 | 0.062 | −0.218 |
| FSIQ | 3242.00 | −3.165 | **0.002** | −0.284 | 2683.50 | −2.665 | **0.008** | −0.271 | 2220.50 | −1.898 | 0.058 | −0.219 | 3001.50 | −0.274 | 0.784 | −0.027 | 303.50 | −0.348 | 0.728 | −0.050 | 493.00 | −0.014 | 0.989 | −0.001 |
| QRI | 3471.00 | −2.022 | **0.043** | −0.182 | 2737.50 | −2.264 | **0.024** | −0.229 | 2110.50 | −3.446 | **0.001** | −0.397 | 2987.00 | −0.384 | 0.701 | −0.038 | 769.00 | −2.064 | **0.039** | −0.298 | 2193.00 | −2.288 | **0.022** | −0.264 |
| AWMI | 27771.0 | −5.543 | **<0.001** | −0.498 | 2659.50 | −2.852 | **0.004** | −0.289 | 2186.00 | −2.388 | **0.017** | −0.275 | 1419.50 | −2.226 | **0.026** | −0.226 | 318.00 | −0.012 | 0.991 | −0.001 | 398.00 | −1.474 | 0.141 | −0.170 |
| CPI | 2893.00 | −4.912 | **<0.001** | −0.441 | 2525.50 | −3.694 | **<0.001** | −0.375 | 2229.00 | −1.781 | 0.075 | −0.206 | 1472.00 | −1.358 | 0.174 | −0.138 | 268.50 | −1.037 | 0.300 | −0.149 | 374.50 | −1.675 | 0.094 | −0.193 |

[a] Mann–Whitney-U; [b] Z-score; [c] asymptotic significance level, bold = *p*-value below 0.050; [d] Pearson correlation coefficient; VCI, Verbal Comprehension Index; WMI, Working Memory Index; PSI, Processing Speed Index; FSIQ, Full Scale IQ; QRI, Quantitative Reasoning Index; AWMI, Auditory Working Memory Index; CPI, Cognitive Proficiency Index.

## Appendix B

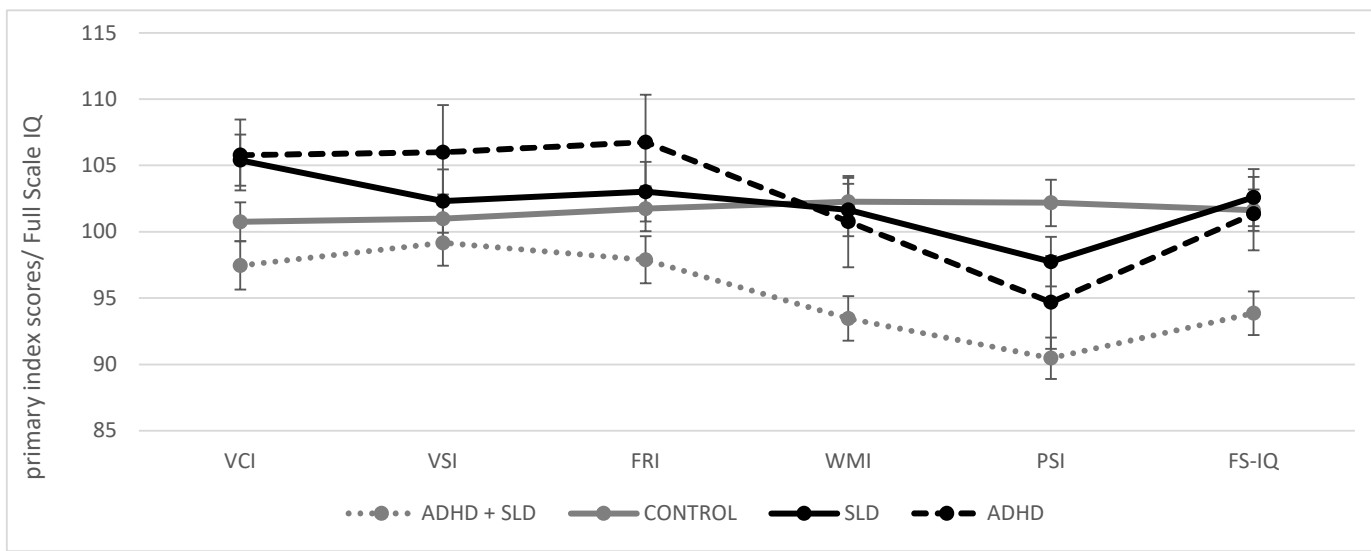

**Figure A1.** Mean scores and standard error of the primary WISC-V index scores and the FSIQ as comparison between groups. Note: VCI, Verbal Comprehension; VSI, Visual Spatial Index; FR, Fluid Reasoning Index; WMI, Working Memory Index; PSI, Processing Speed Index; FSIQ, Full Scale IQ.

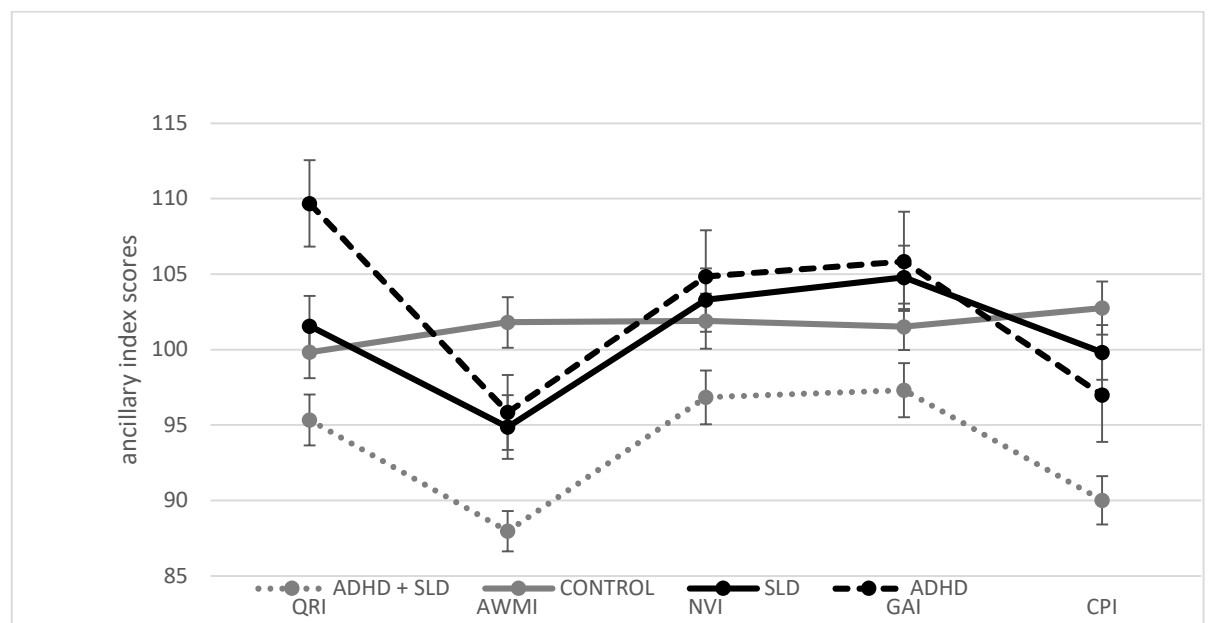

**Figure A2.** Mean scores and standard error of the ancillary WISC-V index scores as comparison between groups. Note: QRI, Quantitative Reasoning Index; AWMI, Auditory Working Memory Index; NVI, Nonverbal Index; GAI, General Ability Index; CPI, Cognitive Proficiency Index.

## Appendix C

**Table A3.** Post-hoc group *t*-test comparisons for scales with significant MANOVA effects.

| WISC-V Index | ADHD + SLD versus CONTROL | | | ADHD + SLD versus SLD | | | SLD versus CONTROL | | |
|---|---|---|---|---|---|---|---|---|---|
| | t(122) | p | d | t(95) | p | d | t(95) | p | d |
| VCI | 1.406 | 0.161 | 0.24 | −2.308 | 0.023 | 0.52 | −1.368 | 0.175 | 0.24 |
| WMI | 3.564 | 0.001 | 0.67 | −3.038 | 0.003 | 0.75 | 0.241 | 0.810 | 0.00 |
| PSI | 4.982 | <0.001 | 0.96 | −2.635 | 0.010 | 0.61 | 1.840 | 0.069 | 0.41 |
| FSIQ | 3.416 | 0.001 | 0.72 | −3.013 | 0.003 | 0.69 | −0.173 | 0.863 | 0.00 |
| AWMI | 6.458 | <0.001 | 1.08 | −2.788 | 0.006 | 0.61 | 2.662 | 0.009 | 0.55 |
| NVI | 1.975 | 0.051 | 0.36 | −2.064 | 0.042 | 0.46 | −0.295 | 0.768 | 0.07 |
| GAI | 1.779 | 0.078 | 0.38 | −2.254 | 0.027 | 0.52 | −0.895 | 0.373 | 0.16 |
| CPI | 5.359 | <0.001 | 0.89 | −3.848 | <0.001 | 0.87 | 1.077 | 0.284 | 0.23 |

VCI, Verbal Comprehension Index; WMI, Working Memory Index; PS, Processing Speed Index; FSIQ, Full Scale IQ; AWMI, Auditory Working Memory Index; NVI, Nonverbal Index; GAI, General Ability Index; CPI, Cognitive Proficiency Index.

## Appendix D

**Table A4.** Mean and standard deviation for the three subtests contributing to the WMI and the AWMI by group and results of Kruskal–Wallis test.

| Subtest | ADHD + SLD n = 62 | | SLD n = 34 | | CONTROL n = 62 | | ADHD n = 13 | | Kruskal-Wallis | | |
|---|---|---|---|---|---|---|---|---|---|---|---|
| | M | SD | M | SD | M | SD | M | SD | $X^2$(df) | | p [a] |
| DS | 7.66 | 2.03 | 9.20 | 2.24 | 10.34 | 2.59 | 9.00 | 2.48 | 33.30 | (3) | <0.001 |
| PS | 10.05 | 3.03 | 11.34 | 2.75 | 10.44 | 3.21 | 11.23 | 2.55 | 4.12 | (3) | 0.249 |
| LN | 7.95 | 4.03 | 8.86 | 2.61 | 10.34 | 2.69 | 9.46 | 1.61 | 25.63 | (3) | <0.001 |

Note. [a] asymptotic significance level. DS, Digit Span; PS, Picture Span; LN, Letter-Number Sequencing. Mann–Whitney-U-Tests were calculated to determine which group differed in the DS and LN subtests. All distributions differed between both groups, Kolmogorov-Smirnov $p < 0.05$. There was a significant difference between the ADHD + SLD and CONTROL group in both DS ($U = 2769.00$; $Z = −5.574$, $p < 0.001$) and LN ($U = 2900.50$; $Z = −4.915$, $p < 0.001$). The ADHD + SLD group differed from the SLD group also in the subtests DS ($U = 2599.00$, $Z = −3.328$, $p = 0.001$) and the subtest LN ($U = 2758.00$, $Z = −2.130$, $p = 0.033$). The ADHD + SLD group differed from the ADHD group only in the LN subtest ($U = 2193.00$, $Z = −2.303$, $p = 0.021$). The SLD Group differed from the CONTROL group only in the subtest LN ($U = 1423.50$; $Z = −2.210$, $p = 0.027$). The ADHD group did not differ significantly from the CONTROL Group in the examined subtests. The ADHD Group also did not differ from the SLD group.

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
