# Peer review of "Cognitive Profiles in the WISC-V of Children with ADHD and Specific Learning Disorders"

_sustainability, doi:10.3390/su13179948_

Round 1

Reviewer 1 Report

Minor Point:

  1. Line 30: The findings in the sixth cited literature indicate that the percentage of SLD is 13.6%. This is significantly different from the 2-8% indicated here, so please reconfirm the value or citation.
  2. Line 31-32: The same problem as above also occurs in [7] - [11]. At the same time, the ratios presented in these literatures have a large discrepancy. It seems that this is not a consensus and the discrepancies in the literature need to be clarified and confirmed.
  3. Line 55: spcific? specific?
  4. Line 56-64: The WISC-V standard is one of the main points in this article. It is possible to describe more about the history of this standard and the differences from the previous version (WISC-IV) here.
  5. Line 72-74: You can refer to this document below.
    WISC-IV Intellectual Profiles in Italian Children With Specific Learning Disorder and Related Impairments in Reading, Written Expression, and Mathematics.
  6. Line 116-122: Split into a second segment. This is because it is another subgroup. The reader will be less confused after the division.  Please confirm again that the total sample tested is 62+35+62=159?
  7. Line 156-160: Please describe more about the statistics in this Table 3, rather than just re-writing the data. Briefly explain the statistical significance of these data.
  8. Line 169 should be merged into Line 168.
  9. Line 172-175: Is the statistical criterion for determining the difference p-value < 0.05 ? It can be graded as 0.05, 0.001, 0.1, so that the meaning can be interpreted more richly.
  10. Line 179-182: The same as Point 8.
  11. According to the cited literature [37], the ADHD discussed in the research can be divided into ADHD-Combined (ADHD-C), ADHD-Inattentive (ADHD-I), comorbid ADHD and DCD, or comorbid ADHD and RD and/or DWE . Can we discuss the differences between the research results and the literature [37] in more depth during the discussion? And the theoretical or practical contributions brought by this research?
  12. There may be specific ADHD groups that are missing from the samples observed in the study, and the characteristics of these groups may be a direction for future research.
  13. Line 20-22: The logical guidance of this passage in the text needs to be described clearly, only to be judged through the literature? It cannot be used as a contribution to this research! The thinking logic of this argument?

Author Response

 We would like to thank the reviewer for taking the time to read our manuscript and provide such extensive and helpful feedback. We have integrated all comments in the updated version of the manuscript and reply in more detail below. We feel that the manuscript improved greatly through the feedback and hope that the reviewer agrees.

Minor Point:

  1. Line 30: The findings in the sixth cited literature indicate that the percentage of SLD is 13.6%. This is significantly different from the 2-8% indicated here, so please reconfirm the value or citation.

#1 Yes, indeed the reference [6] did show a higher prevalence. We excluded it because after looking more closely, we realized that the study did not use standardized test to assess SLD but teacher/parent ratings. We have rechecked the other refences again and will upload the overview of prevalence to which we refer. Thank you for taking the time to checking our refences so thoroughly!

  1. Line 31-32: The same problem as above also occurs in [7] - [11]. At the same time, the ratios presented in these literatures have a large discrepancy. It seems that this is not a consensus and the discrepancies in the literature need to be clarified and confirmed.

#2 We also corrected the numbers from 30-60% to 20-70% and will upload the overview of prevalence that we refer to.

  1. Line 55: spcific? specific?

#3 Thank you for pointing this out. We corrected it.

  1. Line 56-64: The WISC-V standard is one of the main points in this article. It is possible to describe more about the history of this standard and the differences from the previous version (WISC-IV) here.

#4 Thank you for pointing this out. We added the following information: “The WISC-V differs from the previous version of Wechsler intelligence tests, the WISC-IV [25] in several points. The underlying intelligence structure was changed from a four-factor to a five-factor-model that should be adequately represented by the factor structure of the test (see [24], technical manual). On the basis of confirmatory factorial analysis, the Perceptual Reasoning Index was split into the new Visual Spatial Index and the Fluid Reasoning Index. New subtests were added to the test (Figure Weigths, Visual Puzzles and Picture Span). Now only seven subtests contribute to the Full-Scale IQ instead of 10 as in the WISC-IV. The primary Working Memory Index (WMI) now also includes a visual subtest (Picture Span) and an auditory subtest (Digit Span) instead of only auditory subtests as in the WISC-IV, where the WMI was composed of the subtests Digit Span and Letter-Number Sequencing. In the WISC-V these two subtest (Digit Span and Letter-Number Sequencing) are now combined in the ancillary Auditory Working Memory Index.”

  1. Line 72-74: You can refer to this document below.
    WISC-IV Intellectual Profiles in Italian Children With Specific Learning Disorder and Related Impairments in Reading, Written Expression, and Mathematics.

#5 Thanks for the helpful suggestion. We integrated the reference.

  1. Line 116-122: Split into a second segment. This is because it is another subgroup. The reader will be less confused after the division.  Please confirm again that the total sample tested is 62+35+62=159?

#6 Thank you for this suggestion. We split the segment and confirmed again the number of the total sample in the method section (The total number of the tested sample thus is N = 172 (ADHD & SLD = 62, CONTROL = 62, SLD = 35, ADHD = 13).

  1. Line 156-160: Please describe more about the statistics in this Table 3, rather than just re-writing the data. Briefly explain the statistical significance of these data.

#7 Thank you for this suggestion. We decided to omit the redundant statistical values and only determine which group differences were different, plus the information which group differences showed a large effect size. Please note that we added in the Statistical analysis section the information that “The statistical alpha level was set below .05.” We also added the classification of η2 according to Cohen (1988) in this section.  

  1. Line 169 should be merged into Line 168.

#8 Thank you, we decided to split the figure in two figures to make the difference between primary and ancillary index values more clear, now the notes differ. We think that if the notes are merged in one line, the font size is so small so the readability will suffer.

  1. Line 172-175: Is the statistical criterion for determining the difference p-value < 0.05 ? It can be graded as 0.05, 0.001, 0.1, so that the meaning can be interpreted more richly.

#9 Thank you for this suggestion. We added the information that the statistical alpha level was set below .05 in the statistical method section. We added the effect sizes so there is more information about the meaning of the effects. Please note that we refer to the 6th Version of the APA that states: “When reporting p values, report exact p values (e.g., p = .031) to two or three decimal places. However, report p values less than .001 as p < .001. The tradition of reporting p values in the form p < .10, p < .05, p < .01, and so forth, was appropriate in a time when only limited tables of critical values were available.” (p. 114)

  1. Line 179-182: The same as Point 8.

#10 Thank you, we decided to split the figure in two figures to make the difference between primary and ancillary index values more clear, now the notes differ. We think that if the notes are merged in one line, the font size is so small so the readability will suffer.

  1. According to the cited literature [37], the ADHD discussed in the research can be divided into ADHD-Combined (ADHD-C), ADHD-Inattentive (ADHD-I), comorbid ADHD and DCD, or comorbid ADHD and RD and/or DWE . Can we discuss the differences between the research results and the literature [37] in more depth during the discussion? And the theoretical or practical contributions brought by this research?

#11 We had already included the comparison of ADHD RD/DWD vs. ADHD only in the discussion in the first version of the manuscript. We now have an extra paragraph about the exploratory comparison of ADHD+SLD and ADHD in which we name the result of the study of Parke and Colleagues. Since the study did not find differences between the ADHD subtypes (ADHD-C and ADHD-I) and the DSM-V refrains from using the subtypes (and uses predominant presentation instead) we would not like to include the subtype discussion in our manuscript because we fear it would loose focus. Furthermore, we would also like to exclude the discussion about the influence of DCD on the performance in the WISC because this is a whole new discussion to which we cannot add with our manuscript.

  1. There may be specific ADHD groups that are missing from the samples observed in the study, and the characteristics of these groups may be a direction for future research.

#12 Thank you for this important remark. We decided now to include the 13 subjects with ADHD only and added computations with this group in this supplement and also an paragraph in the discussion.

  1. Line 20-22: The logical guidance of this passage in the text needs to be described clearly, only to be judged through the literature? It cannot be used as a contribution to this research! The thinking logic of this argument?

#13 You are absolutely right. We tried to add a sentence about the implications but of course “This highlights the importance of exploring the presence of ADHD in children with SLD to target interventions early and optimally.“ cannot be implied by our results. Therefore, we changed the last sentence of the discussion to:” Hence, the WISC is suited to depict the cognitive strength and weaknesses of an individual child as basis for a targeted intervention rather than as instrument for the differential diagnosis.”

Furthermore, we added a whole paragraph discussing this important implication under “Implications for practice” and extenuated and shorted the other implications which were in fact too much focusing children with ADHD.

Reviewer 2 Report

The article “Cognitive Profiles in the WISC-V of Children with ADHD and specific Learning Disorders” compared three different groups such ADHD+SLD, children with SLD only and a control group. The ADHD alone group is missing this comparison. In the end, the authors concluded that children of “ADHD associated SLD” should be prioritized to intervene compared to SLD without comparing ADHD data. 

The rationale and study question is not strong enough. The prevalence of overlapping the two disorders signifies common control points for both.

The other problem is that the comparison of groups is biased towards control due to the low number of subjects between cases and the large number in control. It may be the case where subjects are not readily available sometimes, however, comparing data with a large number of controls normalize the variation within-group and offer a bias in comparisons across the groups. 

Although working memory (WM) is a highly heritable complex cognitive trait, however, training-induced changes in WM can acquire novel cognitive routines akin to learning a new skill (Journal of Memory and Language Volume 105, April 2019, Pages 19-42). Since WM has an adaptable attribute over years, therefore, SLD groups may be inconsistent. ADHD per se does not necessarily reflect lower intellectual abilities as compared to null, whereas SLD is very specific to the questions asked in this study. Thus, the ADHD+SLD grossly may represent the conditions of SLD concerning the WISC-V test which is being compared with another group of SLD without ADHD. 

Table 1- Group differences in primary education have potentially differed in subject number.

Table 2 – The study is already specified with lower sample size. The demographic dataset suggests an almost equal representation of females across the groups (Table 1). Again, stratifying the subjects into gender in Table 2 does not add any added values to the existing data regarding the questions.

Replace “CG” with “control” enables the clear naming of these groups.

ADHD with 13 children may be considered for comparison. 

The authors stated that the study is limited, as they did not include the ADHD group, however, data from ADHD is required to normalize the basal levels of changes over SLD to evaluate the shared deficits among these groups. The discussion may include data from previous ADHD studies for plausible hypotheses in absence of real-time included in the study.

Author Response

# We would like to thank the reviewer for taking the time to read our manuscript and provide such extensive and helpful feedback. We have integrated all comments in the updated version of the manuscript and reply in more detail below. We think that the manuscript improved greatly through the feedback and hope that the reviewer agrees.

Comments and Suggestions for Authors

The article “Cognitive Profiles in the WISC-V of Children with ADHD and specific Learning Disorders” compared three different groups such ADHD+SLD, children with SLD only and a control group. The ADHD alone group is missing this comparison.

#1 We agree that the missing ADHD only group is a major limitation to our study. We now compare the ADHD group to the other groups using non-parametric procedure to account for the small number of 13 children. Since the analyses with such a small group are only exploratory, we present the results in the appendix but also added a section to the discussion.

In the end, the authors concluded that children of “ADHD associated SLD” should be prioritized to intervene compared to SLD without comparing ADHD data. 

#2 We did not mean to conclude that children with ADHD and SLD should be treated prioritized (before children with SLD or ADHD only). Thanks to your comment we realized that the implication that ADHD should be explored in children with SLD might be important but cannot be drawn from our study. Therefore, we added a new paragraph to the implications discussing that the WISC cannot be used for the differential diagnosis of ADHD vs. SLD vs. ADHD+SLD but rather to display the individual profile of one child’s cognitive strength and weaknesses as a basis for targeted interventions. We extenuated and shorted the other implications which were focusing too much on children with ADHD. We also changed to last sentence of the abstract accordingly.

The rationale and study question is not strong enough. The prevalence of overlapping the two disorders signifies common control points for both.

#3 We feel that although we cannot provide an sufficient ADHD+SLD vs. SLD vs. ADHD vs. CONTROL design we can still add to the literature by providing one of the first studies using the new WISC-V. However, since our sample is not representative and we are missing a sufficient ADHD only group many of our results are not hypothesis testing but rather exploratory and we try to make this clearer in the new version of the manuscript. We hope that we can inspire more profound research (including larger and more representative samples) by our publication and that we can give first helpful insights for practitioners.

The other problem is that the comparison of groups is biased towards control due to the low number of subjects between cases and the large number in control. It may be the case where subjects are not readily available sometimes, however, comparing data with a large number of controls normalize the variation within-group and offer a bias in comparisons across the groups. 

#4 Thank you for pointing this out. Our discussion was focused too much on the comparison of SLD and CONTOL which is not a valid comparison. We changed the structure of the discussion (to focus more on the comparison of ADHD+SLD vs. CONTROL and ADHD+SLD vs. SLD) and added the following sentence: “However, this comparison is limited in its accountability because the control group was matched to the ADHD+SLD group and not to the SLD group and the SLD group has fewer cases compared to the other two groups.”

Although working memory (WM) is a highly heritable complex cognitive trait, however, training-induced changes in WM can acquire novel cognitive routines akin to learning a new skill (Journal of Memory and Language Volume 105, April 2019, Pages 19-42). Since WM has an adaptable attribute over years, therefore, SLD groups may be inconsistent. ADHD per se does not necessarily reflect lower intellectual abilities as compared to null, whereas SLD is very specific to the questions asked in this study. Thus, the ADHD+SLD grossly may represent the conditions of SLD concerning the WISC-V test which is being compared with another group of SLD without ADHD. 

#5 Thank you for this important remark. There are several studies showing that children with ADHD show deficits in at least parts of the working memory (e.g. Hellwig-Brida et al. 2010). A recent study with a German sample showed deficits in the WISC-V Index WMI also in children with ADHD (Pauls et al., 2018). To address this question further, we examined the three subtests that contribute to the Working Memory Index and Auditory Working Memory Index. Here, we also made group comparisons with the small ADHD sample included (see appendix D). The SLD Group differed from the CONTROL group only in the subtest Letter-Number Sequencing. (U = 1423.50; Z = -2.210, p = .027). The ADHD group did not differ significantly from the CONTROL Group in the examined subtests. The ADHD Group also did not differ from the SLD group. We believe that this supports our conclusion that problems accumulate in the ADHD+SLD group and hope that the reviewer agrees.

Table 1- Group differences in primary education have potentially differed in subject number.

#6 Thank you for pointing this out. The reason for the difference in numbers (in the CONTROL group 37 and in the ADHD & SLD group 38 children attended primary school) lies in both the matching of the data and the German School System. We matched for the factors in the order: sex, age, school form and parental educational background. Children with SLD or ADHD (or both as in our study) might have greater struggles in primary school and therefore, in Germany it is not unusual for the children to repeat a grade and remain longer in primary school to have time to catch up possible deficits.

Table 2 – The study is already specified with lower sample size. The demographic dataset suggests an almost equal representation of females across the groups (Table 1). Again, stratifying the subjects into gender in Table 2 does not add any added values to the existing data regarding the questions.

#7 Thank you for pointing this out. The stratification for gender is indeed not helpful and we did change Table 2 accordingly.

Replace “CG” with “control” enables the clear naming of these groups.

#8 Thank you for this notion. We changed it and agree that the naming of the groups is now clearer.

ADHD with 13 children may be considered for comparison. 

#9 As discussed in reply #1 we now included a comparison to the group with ADHD in the supplement and briefly in the discussion.

The authors stated that the study is limited, as they did not include the ADHD group, however, data from ADHD is required to normalize the basal levels of changes over SLD to evaluate the shared deficits among these groups. The discussion may include data from previous ADHD studies for plausible hypotheses in absence of real-time included in the study.

#10 Thanks to your comment we added a paragraph addressing this in the discussion (see 4.3 Exploratory Comparison of ADHD+SLD and ADHD)

Round 2

Reviewer 2 Report

The authors revised their manuscript as per the reviewer's suggestion and made critical structural changes independent of the reviewer's comments. The revised version of the manuscript". Cognitive Profiles in the WISC-V of Children with ADHD and 2 Specific Learning Disorders." has been improved substantially regarding data presentation and additional explanation in the discussion. Thus, the article becomes more attractive despite its inherent limitation with a missing group in the experimental design.